# Graphene-PEDOT: PSS Humidity Sensors for High Sensitive, Low-Cost, Highly-Reliable, Flexible, and Printed Electronics

**DOI:** 10.3390/ma12213477

**Published:** 2019-10-24

**Authors:** Vasiliy I. Popov, Igor A. Kotin, Nadezhda A. Nebogatikova, Svetlana A. Smagulova, Irina V. Antonova

**Affiliations:** 1Institute of Physics and Technologies, North-Eastern Federal University, Yakutsk 677027, Russia; volts@mail.ru (V.I.P.);; 2Rzhanov Institute of Semiconductor Physics, Novosibirsk 630090, Russia; kotin@isp.nsc.ru (I.A.K.);; 3Department of Physics, Novosibirsk State University, Novosibirsk 630090, Russia; 4Department of Semiconductor Devices and Microelectronics, Novosibirsk State Technical University, Novosibirsk 630073, Russia

**Keywords:** humidity sensors, graphene: PEDOT: PSS composite, printed structures, response, flexible test devices

## Abstract

A comparison of the structure and sensitivity of humidity sensors prepared from graphene (G)-PEDOT: PSS (poly (3,4-ethylenedioxythiophene)) composite material on flexible and solid substrates is performed. Upon an increase in humidity, the G: PEDOT: PSS composite films ensure a response (a linear increase in resistance versus humidity) up to 220% without restrictions typical of sensors fabricated from PEDOT: PSS. It was found that the response of the examined sensors depends not only on the composition of the layer and on its thickness but, also, on the substrate used. The capability of flexible substrates to absorb the liquid component of the ink used to print the sensors markedly alters the structure of the film, making it more porous; as a result, the response to moisture increases. However, in the case of using paper, a hysteresis of resistance occurs during an increase or decrease of humidity; that hysteresis is associated with the capability of such substrates to absorb moisture and transfer it to the sensing layer of the sensor. A study of the properties of G: PEDOT: PSS films and test device structures under deformation showed that when the G: PEDOT: PSS films or structures are bent to a bending radius of 3 mm (1.5% strain), the properties of those films and structures remain unchanged. This result makes the composite humidity sensors based on G: PEDOT: PSS films promising devices for use in flexible and printed electronics.

## 1. Introduction

In recent years, graphene-based gas or vapor sensors have attracted much attention due to their unique sensing performance, room-temperature operating conditions, a variety of feasible designs, and tremendous application perspectives. Over the past few decades, the rapid development of graphene-based sensors for various applications has been observed. Humidity measurement is one of the most significant issues in the fields of medical diagnostics, environmental protection, industrial manufacture, agriculture, etc. Graphene demonstrates great potential for application in resistive humidity sensors [1]. The sensitivity of a sensor prepared from chemical vapor deposition (CVD) grown graphene is normal rather than moderate (resistance change <2%) [2,3]. A transition to bi-graphene or multi-graphene increases the sensitivity of the sensor up to ~10%–30% and, in some cases, even up to ~90% depending on the defect inventory of the material since defects act as the centers of water adsorption on the sensor surface [4,5,6,7].

One of the organic polymers, poly (3,4-ethylenedioxythiophene) doped with poly (styrene sulfonate) (PEDOT: PSS), is a material most often used in humidity-sensor applications [8]. PEDOT: PSS is a polymer exhibiting the highest conductivity among other organic materials [9]. Resistive humidity sensors based on PEDOT: PSS demonstrate a high sensitivity (of 430% at humidity changes ranging from 40% to 90%) and a linear dependence of their response on humidity. However, at a high humidity of more than 80% such sensors lose their functional properties due to the saturation of the film with water or due to the formation of a water layer on the surface of the film at humidity exceeding 80%, with the latter process leading to a decrease of resistance and to the necessity of a long recovery of the sensor [8]. Therefore, PEDOT: PSS is widely used in various composites in combination with other organic materials such as polyvinyl alcohol [10], ZnSnO_3_ or Fe_2_O_3_ nanoparticles [11,12], graphene quantum dots, and silver nanowires [13]. The use of composite materials allows one to achieve high sensitivity (up to ~100%) of the sensors; however, the nonlinearity of characteristics substantially restricts the area of their application.

Particular interest in humidity sensors based on graphene and PEDOT: PSS is due to the possibility of using those materials in flexible, stretchable, and wearable electronics as well as in the printing technology for creating device structures based on such sensors. For an instant, printed PEDOT: PSS electrodes showed good flexibility in several types of flexibility tests, including outer/inner bending, twisting, and stretching [14]. The high-performance silver grid/ PEDOT: PSS hybrid transparent films exhibit promising features for various emerging flexible electronics and optoelectronic devices. [15]. Smart textiles or woven electronics and optoelectronics, attained by intertwined fibers with complementary functions, are the emerging and most ambitious technological and scientific frontier which attained for intertwined fibers with complementary functions due to graphene covering [16]. The fabric-enabled pixels for displays and position sensitive functions are a gateway for novel electronic skin, wearable electronic, and smart textile applications [16,17].

In the present study, we examined the properties of films prepared from the G: PEDOT: PSS composite versus the composition of the material and as dependent on the solid and flexible substrates used. A composition ensuring the maximum sensitivity of the film to water adsorption was identified. It was found that the use of flexible substrates, due to their capability to quickly absorb the liquid components of the solution, leads to a more porous structure of the films, thus making the response of such a humidity sensor increase substantially. An analysis of the properties of such films under deformation proved the films show promise for use in flexible electronics.

## 2. Preparation of Composite Films and Test Device Structures

A graphene suspension with a graphene content of ~1 mg/ml was used as the starting material in the present study. The suspension was obtained by electrochemical exfoliation of highly oriented pyrolytic graphite, followed by the sonication, centrifugation, and filtration of obtained particles on a track membrane with a pore diameter of 1.2 μm [18]. The second component of the composite material was a 1% aqueous solution of PEDOT: PSS. The data on the used compositions of the composite are summarized in Table 1. In obtaining the films and device structures, the screen printing was used. For suppressing the coffee-ring effect during printing, ethylene glycol was added to the final composition. As a result, two types of structures were fabricated: (1) strips 1.5 mm wide whose thickness varied from 300 nm to 7 μm and (2) zigzag test structures about 3 μm thick with line widths of 0.3 and 1 mm (see the insets in Figure 5). Since identical suspension volumes were used to create a series of structures, the thickness of the resulting films in the structures was approximately the same within one series of specimens or it changed in accordance with the suspension volume used. The film thicknesses determined using SEM, AFM, and optical microscopy measurements are indicated in Table 1. In preparation of the samples, SiO_2_/Si wafers, Kapton polyimide films, polyethylene terephthalate (PET) with additional hydrophilic adhesion layer on the surface, and ordinary paper were used as the substrates.

## 3. Results

### 3.1. The Structure of the Graphene: PEDOT: PSS Composite Films

Figure 1 shows SEM images of the surface of the films printed from a suspension of graphene, PEDOT: PSS, and C1a1 composite on the surface of SiO_2_/Si substrates. In the case of PEDOT: PSS films, the removal of part of the substrate has allowed us to obtain an image of a free film. Raman spectra of the films are shown in Figure 1d. In the case of the composite, the spectra present the sum of peaks characteristic of graphene and PEDOT: PSS.

AFM images of the surface of the films obtained using a suspension of graphene and PEDOT: PSS, and AFM images of the surface of the C1a1 and C1c structures with one and the same film composition yet printed on different substrates are shown in Figure 2. Evidently, the PEDOT: PSS film is more porous. It is the property that ensures a high response of such films to absorb water molecules. It is important that, with the same film composition, the structure of the films, as it turned out, could change fundamentally depending on the substrate used. In the case of paper, the film proved to be much more porous (compare Figure 2c,d).

### 3.2. The Response of the Humidity Sensors

Figure 3a shows the relative change in the resistance of a G: PEDOT: PSS composite film as a function of time during a 30% to 90% change of air humidity in the measurement chamber. It is seen that in the case of pure PEDOT: PSS (composition 0:1), there occurs almost no recovery of resistance due to the saturation of the material with water molecules (this circumstance makes the relaxation process long). In the case of an increased graphene content, the rate of recovery increases, and for the composition of graphene: PEDOT: PSS 1:(1–2), it attains a maximum possible value. Later, the rate of increase/decrease in humidity is limited by the inertia of the chamber.

The response of the structures with different compositions observed on the humidity change to 90% is shown in Figure 3b. It is seen that the dependence of the response on the composition of the suspension from which the film was formed demonstrates a maximum. That maximum refers to the composition of 35%–50% of graphene. Yet, the most important consequence of the data shown in Figure 3b is a strong effect due to the substrates on which the film was printed. Evidently, the transition to flexible substrates, capable of absorbing part or all the liquid applied onto the printed layer, drastically changes the film structure. In our case, it is the paper and the PET with an additional hydrophilic adhesive layer on the surface. This feature of flexible substrates may significantly affect the sensitivity of the sensors. In the case of paper used as the substrate, the absorption of the liquid component occurred most rapidly; this, as it follows from the data of Figure 2, led to the formation of a porous film structure yielding a maximum response to moisture.

No less important parameters for the response to moisture is the film thickness. Among the structures fabricated on SiO_2_/Si substrates, the C1a2 structure with a minimum film thickness of 300 nm exhibited a maximum response of 27%–60%, and at a film thickness of 3.5 μm, and that response dropped to 15%–35%. In the case of relatively thick G: PEDOT: PSS composite films formed on paper (6.5 to 7.5 μm thick), the film thickness was no longer an obstacle for reaching a pronounced response due to the porosity of the film formed on paper. Moreover, a decrease in film thickness to 3 μm led to a decrease in response to 72%.

Figure 4 shows the curves of the resistance of the samples obtained from suspensions with the same composition (1:1) on different substrates; those curves were recorded during a two-sided sweep of humidity. Despite the micron thicknesses of the sensing layer, due to its porosity, not only high response but, also, almost linear dependences of the resistance on the humidity and the absence of hysteresis were obtained. The relatively weak hysteresis observed for the structure on paper was most likely associated with the capability of the substrate to absorb moisture and then give it back.

In general, it should be noted that the rate of change of the resistance is determined, first of all, by the inertia of the measurement chamber. The inset in Figure 4a shows the changes in the resistance and humidity due to a sharp opening and closing of the chamber. Both composite sensors and, also, the HIH4000 industrial capacitive sensor, with the fast operation, displayed this event on the same time scale.

### 3.3. Testing of the Functional Properties of the Humidity Sensors under Mechanical Deformations

The study of the changes in the resistance of the structures during deformation was carried out by applying bending deformations to these structures. Depending on the bending direction, tensile (positive) or compressive (negative) strains could be generated. The strain estimated from the bending radius ***r*** at the substrate thickness ***t*** = 100 μm and film thickness ***d*** amounts to *=***(*t**+**d*)/*2r*** [18]. On a change of the bending radius to 3 mm, the strain in the structures on PET films increased to 1.5%. Figure 5 shows the resistance changes for a strip printed on PET from a G: PEDOT: PSS suspension and one of the fabricated zigzag test structures. In the first case, a relatively weak increase in resistance under tensile deformations and an almost unchanged resistance under compressive deformations were observed. The scatter of resistance values during repeated measurements was most likely associated with a poor reproducibility of the position of the sample when the sample was mounted in the holder; this circumstance could lead to a change in the current paths in the film and to a scatter of resistance values [19,20]. In the case of measuring the zigzag test structure (Figure 5b), the resistance during deformation and during repeated measurements remained almost unchanged. The effect due to the shape of the tested structure on the maximum possible deformations is well known [21,22,23,24] but, in our case, the creation of a test structure with a zigzag shape led to a profound reduction in the scatter of resistance values.

Figure 6 shows the results of measurements of the test structures of humidity sensors in the deformed state. It is seen that the bending of structures in the chamber during exposing them to moisture practically did not change at the implemented bending radii (the minimum bending radius was 3 cm, which value corresponds to a tensile strain of 1.5%).

## 4. Discussion

The obtained experimental results can be summarized as follows. 

(a) The content of the composite was identified at which the response of the sensor to humidity was maximum. This is a composite in which a suspension of graphene and PEDOT: PSS is in a proportion of 1:(1–2). It is shown that the response of the system is unambiguous, the resistance increases by increasing the humidity up to 95%. Film recovery occurs on the same time scales in the entire range of humidity. Estimations of the 1:1 G: PEDOT: PSS film weight content after drying of printed layer gives the value of 10% addition of PEDOT: PSS to graphene and 5% in the case of 1:2. In our paper [7] we have demonstrated that edge defects determine the graphene sensor response. Addition of the relatively small mass value of PEDOT: PSS to graphene creates better conditions for humidity sensitivity due to better access to individual flakes and their edges in comparison with a monolithic film formed from graphene. Good water absorption by PEDOT: PSS also leads to a higher response.

(b) The use of flexible substrates capable of absorbing the liquid ink component permits the formation of sensor elements with high porosity. As a result, the relatively large thickness of such films does not lead to a loss of layer sensitivity.

(c) As it could be expected, in the case of using a rigid SiO_2_/Si substrate a maximum sensor response was observed for the thinnest film.

The used G: PEDOT: PSS composites and test structures possess properties promising for the use of such humidity sensors in flexible and printed electronics: When the structures are bent to a bending radius of 3 mm (1.5% deformation), their properties remain almost unchanged. A pronounced reduction of the scatter of resistance values was obtained for zigzag-shaped test structures.

The bending-induced effects in flexible electronics have emerged as a relatively new experimental and theoretical research area. The optimum device design for flexible electronics requires a comprehensive understanding and accurate modeling of strain-induced effects in sensor devices [25]. The shape of the substrate, its parameters, and the resultant distribution of strain during bending can be expected to exert an influence on the responses of the devices formed on flexible substrates. Uniaxial deformation occurs when an applied bending moment deforms the specimen along an in-plane axis. The use of thin substrates can lead to a distortion of the surface because of the various types of occurred deformations such as the tensile, compressive, shearing, and torsional ones. Those deformations can be quantified in terms of stresses and strains having components such as uniaxial, biaxial, and torsion bending deformations. Namely, this type of thin substrates entails a scatter of resistance values shown in the present study in Figure 5a. A periodic serpentine shape provides a regular contribution of the various components of the deformations, thus reducing the spread of resistance values (Figure 5b). Generally, the periodic serpentine shape can be engineered to accommodate the enhanced elastic strain along some selected dimensions and to support the biaxial, radial, and other deformation modes. In addition, the choice of topologies ranging from lines to a serpentine shape is capable of tailoring those topologies to particular electronic applications.

## 5. Conclusions

The properties of graphene: PEDOT: PSS composite films are compared as a material for resistive humidity sensors for moisture range 30%–95% fabricated on flexible and rigid substrates. It is shown that the resistance increases almost linearly depending on the environmental humidity. The composite films G: PEDOT: PSS provide the maximum response (increase in resistance) up to 220% for film content 1:(1–2) (10–5 weight % of PEDOT: PSS in graphene film) when exposed to a humid environment without the restrictions typical of humidity sensors prepared from PEDOT: PSS (80% RH). It is found that the response of the sensor depends not only on the composition of the layer but, also, on its thickness, film structure, and substrate used. Formation of the porous film, for instance, due to the capability of flexible substrates to absorb the liquid component of the ink, used to print the sensing layer, leads to increase the response to humidity. However, in the case of using paper as a substrate, a hysteresis occurs in the resistance value under a two-sided variation of humidity. This hysteresis is associated with the capability of such substrates to absorb water. The measurements of the resistivity of G: PEDOT: PSS films under bending deformation showed that down to a bending radius of 3 mm (1.5% deformation), the resistance of the structures remained almost unchanged. Moreover, the transition from strips to zigzag structures reduces the scatter of resistance values observed during multiple measurements. The response of the test device structures also remained almost unchanged for a tensile bending radius from 3 cm to 3 mm. Thus, used composite material and humidity sensors prepared from it are highly promising for use in flexible and printed electronics.

## Figures and Tables

**Figure 1 materials-12-03477-f001:**
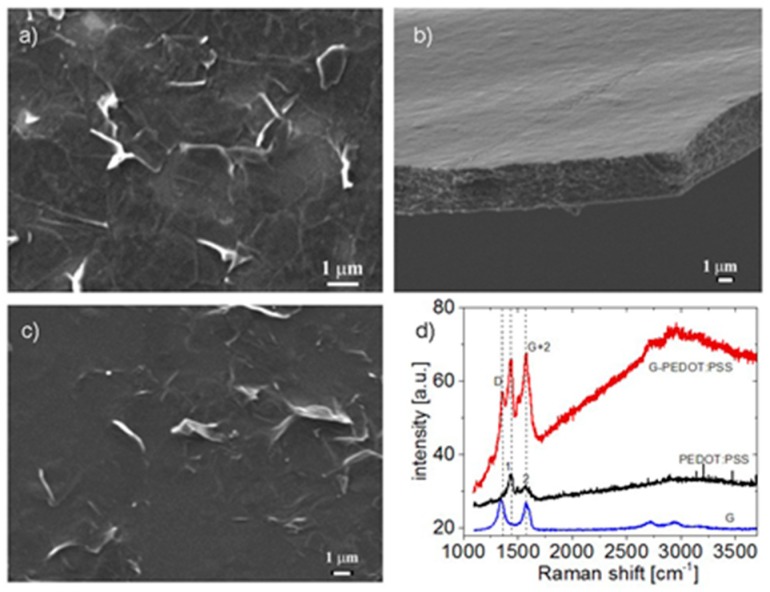
SEM images of the surface of the films printed from a suspension of graphene (**a**), PEDOT: PSS (**b**), and C1a1 composite (**c**) on the surface of SiO_2_/Si substrates. Raman spectra of graphene, PEDOT: PSS, and C1a1 composite films printed on SiO_2_/Si substrates (**d**).

**Figure 2 materials-12-03477-f002:**
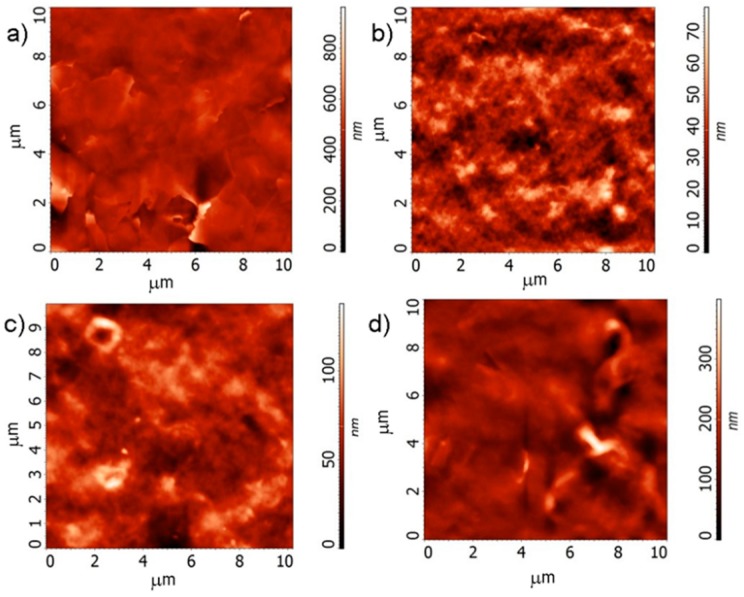
AFM images of the surface of the films printed from a suspension of graphene (**a**), PEDOT: PSS (**b**), (**c**,**d**), and C1c and C1a1 composites (**c** and **d**, respectively).

**Figure 3 materials-12-03477-f003:**
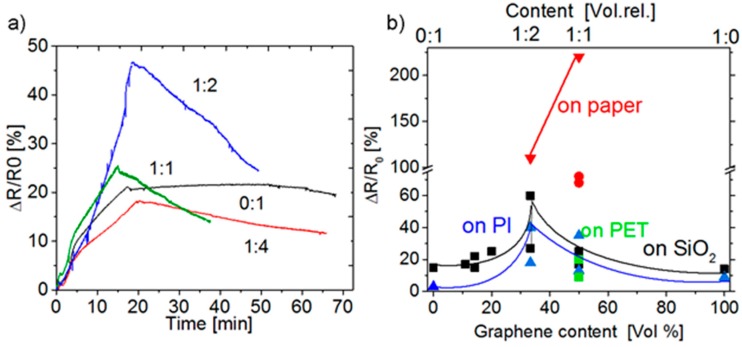
(**a**) The relative change in the resistance of a G: PEDOT: PSS composite film as a function of time during the variation of air humidity in the measurement chamber from 30% to 90%. The humidity was changed linearly with time. The content of the composite layer printed using the ink prepared from a suspension of graphene and PEDOT: PSS is shown at the curves as a parameter. Humidity was measured with an HIH4000 industrial humidity sensor. (**b**) The dependence of the response of the structures printed on different substrates on the composition of the film. The thickness of the films printed on paper was ~7 and 3 μm (triangles and circles, respectively), and for the films on SiO_2_/Si and PI, about 3.5 μm.

**Figure 4 materials-12-03477-f004:**
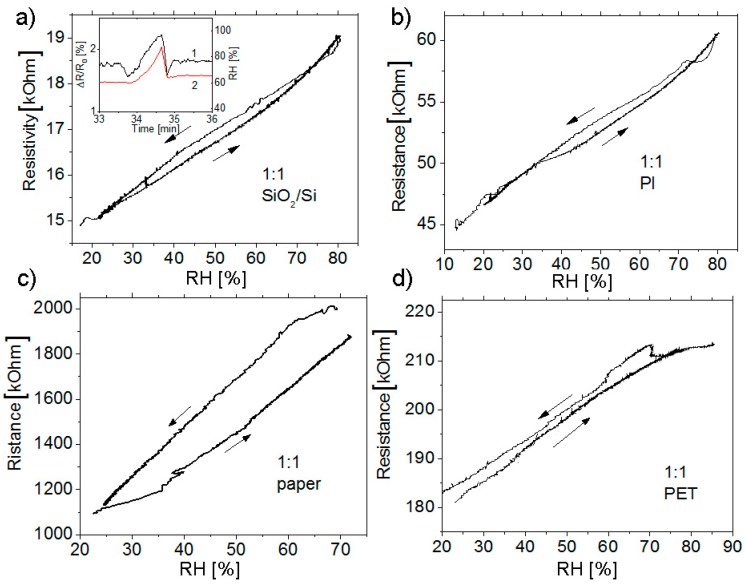
Dependences of resistance on the humidity for C1a1 and C1b strips (**a**,**b**) and for test structures (**c**,**d**) grown on different substrates. The resistance was measured on changing the humidity in two directions. The composite graphene layer is PEDOT: PSS 1:1. The humidity was measured with the HIH4000 industrial humidity sensor. Inset (**a**): the relative change in the resistance of the C1a1 film during a sharp decrease of humidity in the chamber. Red line (curve 2) shows the data taken from the HIH4000 humidity sensor.

**Figure 5 materials-12-03477-f005:**
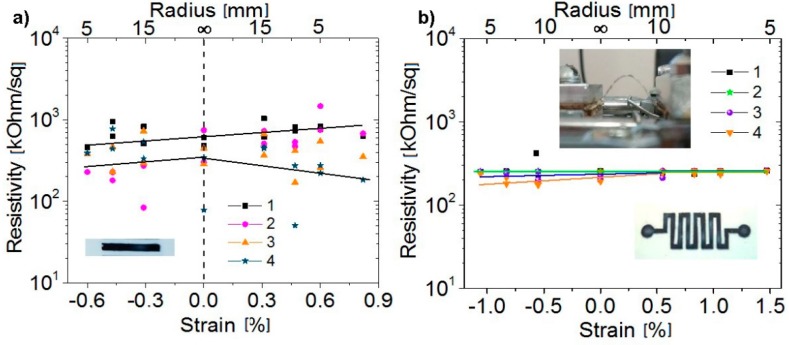
Variation of the resistance of a strip printed on PET (polyethylene terephthalate) from the G: PEDOT: PSS composite suspension (**a**) and that of a test structure (**b**). In both cases, four cycles of repeated measurements were made. The insets show photographs of the structures studied.

**Figure 6 materials-12-03477-f006:**
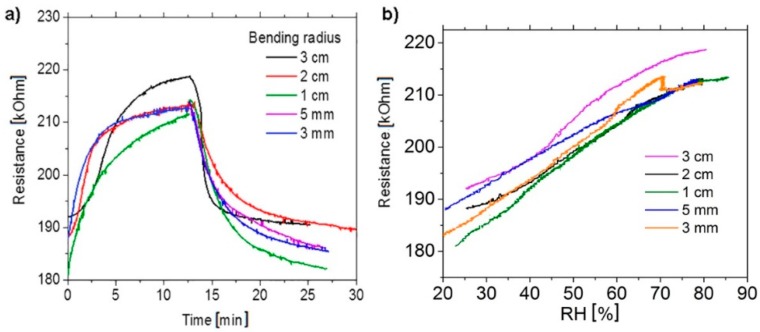
(**a**) Relative change of the resistance of the test structure (see the inset in Figure 6c) printed from G: PEDOT: PSS on a PET substrate as a function of time during the variation of air humidity in the measuring chamber for different tensile deformations of the structure arising from bending. The bending radius is given as a parameter. (**b**) Variation of the resistance of one and the same test structure when sweeping the humidity in two directions versus the bending radius of the structure.

**Table 1 materials-12-03477-t001:** Nomenclature and characteristics of the structures prepared from the graphene: PEDOT: PSS composite material. R/R_o_ is the response of humidity sensor, R/R_o_ = (R(high humidity)−R_o_(low humidity))/R_o_ [%]. High humidity was equal to 80%–90%, low humidity was equal to 20%–30%.

Sample	Content, the Volumetric Ratio Graphene: PEDOT: PSS	Used Substrate	Film Thickness, m	Response R/Ro, %
Graphene	1:0	SiO_2_/Si	0.6 ± 0.2	14
PEDOT: PSS	0:1	SiO_2_/Si	3.5 ± 0.5	15
C1a1 C1a2 C1b C1c	1:1	SiO_2_/Si SiO_2_/Si PI, Kapton paper	3.5 ± 0.5 0.3 ± 0.05 3.5 ± 0.5 7.0 ± 0.5	15–35 27–60 18–40 110
C2a C2b C2c	1:2	SiO_2_/Si PI, Kapton paper	3.5 ± 0.5 3.4 ± 0.5 7.0 ± 0.5	17–25 15–35 220
C3	1:3	SiO_2_/Si	3.5 ± 0.5	24
C4	1:4	SiO_2_/Si	3.5 ± 0.5	15–22
C5	1:8	SiO_2_/Si	3.5 ± 0.5	17
Test 1	1:1	Paper	3.0 ± 0.5	58–72
Test 2	1:1	PET	3.0 ± 0.5	10–20

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
