# Peer review of "Graphene-PEDOT: PSS Humidity Sensors for High Sensitive, Low-Cost, Highly-Reliable, Flexible, and Printed Electronics"

_materials, 2019, doi:10.3390/ma12213477_

Round 1

Reviewer 1 Report

In this manuscript, the authors prepared graphene-PEDOT:PSS composite films with various graphene percentage and on different substrates. They compared the performance of these films as humidity sensors and also studied their resistance change during deformation. However, I can’t recommend this manuscript to be published at this stage before the authors address the following comments properly:

In table 1, there is a column about ‘Response R/Ro, %’. What is that? How did the authors get these numbers? The authors claim the PEDOT:PSS film is more porous based on AFM measurement in Fig 2. I would suggest they measure the surface roughness to compare these films quantitatively. ‘Fig 3a shows the relative change in the resistance of a G:PEDOT:PSS composite film as a function of time during a 30-to-90% change of air humidity in the measurement chamber.’ Did the humidity change linearly with time? What was initial humidity and what was the humidity at 70 min? If the humidity increased with time, how did the authors measure ‘the rate of recovery’? In Fig 3a, why do the curves of 1:1 and 1:2 end earlier? The curve of 1:1 stops less than 40min, and the curve of 1:2 stops at 50min. In Fig 3b, there are 4 results for composite films on paper. What is the difference between the triangle point and the circle point? Why does the 1:1 triangle point have much higher intensity than the circles? To explain why the composite film on paper has the highest sensitivity, the authors claimed that the paper can absorb liquid that can further change the film structure. However, they didn’t provide any direct experimental evidence. Can the authors characterize the film on paper when the humidity is high? Moreover, can the authors explain why 1:1 G:PEDOT:PSS has a better sensitivity than 1:2 G:PEDOT:PSS? On page 5, line 137, the authors wrote ‘evidently, the transition to flexible substrates, capable of absorbing part or all the liquid applied onto the printed layer drastically changes the film structure’. However, PI and PET also are flexible substrates. But they don’t absorb liquid. The authors cited Figure 6 in the article, but I couldn’t find it. In Fig. 5 a), only two curves can be observed. Where are the other two curves? The abstract and the conclusions of this manuscript are exactly the same! Please rephrase.

Author Response

List of corrections and answer to reviewers

Thank you very much for the help in the improvement of our manuscript. Reviewer comments are very useful. I hope now our manuscript is written more clearly now. All corrections are marked in the text given below by yellow.  In addition, we have corrected the English language ( this changes are not marked in the text).

Reviewer 1

In table 1, there is a column about ‘Response R/Ro, %’. What is that? How did the authors get these numbers?

The response of resistive humidity sensor is definite as DR/Ro = ( R(high humidity) – Ro(low humidity))/ Ro [%]. We have added this definition to the text. High humidity was equal to 80-90%. Low humidity was equal to ~20-30%.

The authors claim the PEDOT:PSS film is more porous based on AFM measurement in Fig 2. I

would suggest they measure the surface roughness to compare these films quantitatively.

We have check surface roughness and have found that roughness does not match the porosity of the film. Maximum roughness has graphene film when graphene flakes stick out on the surface.

‘Fig 3a shows the relative change in the resistance of a G:PEDOT:PSS composite film as a function of time during a 30-to-90% change of air humidity in the measurement chamber.’ Did the humidity change linearly with time? What was initial humidity and what was the humidity at 70 min? If the humidity increased with time, how did the authors measure ‘the rate of recovery’?

Initial humidity is equal to ~30%. The humidity was changed linearly with time up to humidity 80-90%. The curve corresponded to PEDOT:PSS has a maximum value of about 80%. Rate of recovery is the slope of curve R(t) = DR/Dt.

In Fig 3a, why do the curves of 1:1 and 1:2 end earlier? The curve of 1:1 stops less than 40min, and the curve of 1:2 stops at 50min. In Fig 3b, there are 4 results for composite films on paper. What is the difference between the triangle point and the circle point? Why does the 1:1 triangle point have much higher intensity than the circles?

We have used different rates of humidity increase and did not found pronounced changes in the response. The thickness of the films printed on paper was ~7 and 3 μm (triangles and circles, respectively). The decrease in the thickness of the porous film leads to a decrease in response.

To explain why the composite film on paper has the highest sensitivity, the authors claimed that the paper can absorb liquid that can further change the film structure. However, they didn’t provide any direct experimental evidence. Can the authors characterize the film on paper when the humidity is high?

We have related to the possibility of paper to absorb the liquid from inks as the main factor to the creation of porous film with a high relation of the surface/volume (Fig2b ). The porous film structure is clearly seen in the AFM image. The surface roughness is not reflect the porous film structure  due to the relatively high suface relief. The hysteresis in the case of two side humidity changes is clearly observed (see, Fig.4c ). So, we have found that  the AFM images are the main  experimental evidence of  the porous film structure.

Moreover, can the authors explain why 1:1 G:PEDOT:PSS has a better sensitivity than 1:2 G:PEDOT:PSS?

Estimations of the 1:1 G:PEDOT:PSS film weight content after drying of printed layer gives the value of 10% addition of PEDOT:PSS to graphene and 5% in the case of 1:2.  In our paper [7] we have demonstrated that edge defects determine the graphene sensor response.  Addition of the relatively small mass value of PEDOT:PSS to graphene creates better conditions for humidity sensitivity due to better access to individual flakes and their edges in comparison with a monolithic film formed from graphene. Good water absorption by PEDOT:PSS also leads to a higher response.

On page 5, line 137, the authors wrote ‘evidently, the transition to flexible substrates, capable of absorbing part or all the liquid applied onto the printed layer drastically changes the film hydrofilling structure’. However, PI and PET also are flexible substrates. But they don’t absorb liquid.

We have corrected the phrase given above.

Evidently, the transition to such flexible substrates as paper, capable of absorbing part or all the liquid applied onto the printed layer drastically changes the film structure’

As it conserved to PI, the response of the sensors on PI is very low due to the dense structure of the G:PEDOT:PSS film. PI film (Kapton) has no additional adhesion layer. PET film has an additional hydrofillic layer on the surface to provide better adhesion. This layer provides water absorption from the printed layer and porous structure.

The authors cited Figure 6 in the article, but I couldn’t find it. In Fig. 5 a), only two curves can be observed. Where are the other two curves?

I did not understand why the referee did not find the Fig.6. I have checked the pdf file submitted to the journal. There is a fig.6.

The abstract and the conclusions of this manuscript are exactly the same!

The abstract and conclusions are corrected.

Reviewer 2 Report

This manuscript presents a detailed study of humidity sensors based on graphene:PEDOT-PSS. The linear response range of the detectors is greatly enhanced by the addition of graphene. These sensors are mechanically flexible and potentially of interest to a wide community working on wearable electronics. I strongly encourage the authors to cite in the introduction to their manuscript the most recent advances in the field of humidity sensors with graphene and wearable technologies such as textile electronics [npj Nature Flexible Electronics, volume 2, pages 25 (2018) DOI:10.1038/s41528-018-0040-2] and with a demonstrate compatibility with roll-to-roll technologies [E. Torres Alonso et al., Advanced Science 6, 15 (2019)  https://doi.org/10.1002/advs.201802318 ]. By linking the manuscript of humidity sensors based on graphene:PEDOT-PSS to the most recent advances in textile graphene electronics and roll-to-roll humidity sensors the authors will certainly benefit of an increase in the visibility of their work.

Author Response

Thanks to Referee we have added the most recent advances in the field of humidity sensors with graphene: We gave added few references in the introduction including recommended ones.

Round 2

Reviewer 1 Report

In this response letter, the authors answered most of my questions and revised the manuscript, except this question ‘In Fig. 5 a), only two curves can be observed. Where are the other two curves?’

In addition to this, I highly encourage the authors to integrate their answers to my questions into the manuscript. Because the readers may have the same questions as me.

Another thing to remind is that some of the symbols don’t display correctly in the revised manuscript, such as the delta symbol, and the equation in section 3.3.

Author Response

I have added ref 3 and 14-17 in the Introduction. These references include the suggested ones. Now I have find that Torres Alonso (first author) is most likely is family name and I corrected this name in Ref 3 and 17. Also I have corrected some numbers of references in the text.